



# Baseline data for monitoring geomorphological effects of glacier lake outburst flood: A very high-resolution image and GIS datasets of the distal part of the Zackenberg River, northeast Greenland

Aleksandra M. Tomczyk[1], Marek W. Ewertowski[1]

[1]Faculty of Geographical and Geological Sciences, Adam Mickiewicz University, Poznań, 61-680, Poland

*Correspondence to*: Aleksandra M. Tomczyk (alto@amu.edu.pl)

**Abstract.**

The Arctic regions experience intense transformations, such that efficient methods are needed to monitor and understand Arctic landscape changes in response to climate warming and low-frequency high-magnitude events. One example of such events,

capable of causing serious landscape changes, is glacier lake outburst floods. On 6 August 2017, a flood event related to glacial lake outburst affected the Zackenberg River (NE Greenland). Here, we provided a very high-resolution dataset representing unique time-series of data captured immediately before (5 August 2017), during (6 August 2017), and after (8 August 2017) the flood. Our dataset covers a 2.1-km-long distal section of the Zackenberg River. The available files comprise: (1) unprocessed images captured using an unmanned aerial vehicle (UAV): https://doi.org/10.5281/zenodo.4495282 (Tomczyk

and Ewertowski, 2021a); and (2) results of structure-from-motion (SfM) processing (orthomosaics, digital elevation models, and hillshade models in a raster format), uncertainty assessments (precision maps) and effects of geomorphological mapping in vector formats: https://doi.org/10.5281/zenodo.4498296 (Tomczyk and Ewertowski, 2021b). Potential applications of the presented dataset include: (1) assessment and quantification of landscape changes as an immediate result of glacier lake outburst flood; (2) long-term monitoring of high-Arctic river valley development (in conjunction with other datasets); (3)

establishing a baseline for quantification of geomorphological impacts of future glacier lake outburst floods; (4) assessment of geohazards related to bank erosion and debris flow development (hazards for research station infrastructure – station buildings and bridge); (5) monitoring of permafrost degradation; and (6) modelling flood impacts on river ecosystem, transport capacity, and channel stability.

## 1 Introduction

Long-term riverscape evolution is the effect of an interplay between "normal" (i.e., low-magnitude, high-frequency geomorphological work) and extreme processes (i.e., high-magnitude, low-frequency events) (cf. Death et al., 2015; Garcia-Castellanos and O'Connor, 2018). One of the critical issues in fluvial geomorphology is the quantification of geomorphological effects caused by both groups of processes that affect river channel morphology and functioning. The problem is that



catastrophic events are hard to predict, such that our ability to collect qualitative data about their direct impact is limited, and
yet this knowledge is crucial for river monitoring and modelling (Tamminga et al., 2015a; Tamminga et al., 2015b).

Among the most severe flood-related extreme events are glacier lake outburst floods (GLOFs), usually related to a sudden
release of water stored in ice-dammed or moraine-dammed lakes, and commonly occur in modern glacierized mountain areas
(Russell and Arnott, 2003; Russell et al., 2007; Moore et al., 2009; Watanabe et al., 2009; Chen et al., 2010; Cui et al., 2010;
Jansky et al., 2010; Osti et al., 2011; Wang et al., 2011; Raj and Kumar, 2012; Iribarren et al., 2015; Emmer, 2017; Harrison
et al., 2018; Nie et al., 2018; Carrivick and Tweed, 2019). The direct cause of the water release is usually related to: (1) increase
in water level in subglacial lakes, causing ice flotation and breaching of the ice dam (Tweed and Russell, 1999; Roberts et al.,
2003); (2) breaching of the moraine dam (Watanabe and Rothacher, 1996; Reynolds, 1998; Westoby et al., 2014); (3) increase
in the amount of meltwater due to the explosion of subglacial volcanoes (Carrivick et al., 2004; Russell et al., 2010).

GLOFs can vary in size and frequency, and yet such flood events can significantly impact river morphology, as they often far
exceed the potential maximum of meteorological floods (Desloges and Church, 1992; Cook et al., 2018; Garcia-Castellanos
and O'Connor, 2018). As such, the documentation of the geomorphological records of such events is essential for the prediction
and management of future transformations in the context of ongoing climate changes (Nardi and Rinaldi, 2015; Carrivick and
Tweed, 2016) that can cause an intensification of these flood events (Reynolds, 1998; Harrison et al., 2006; Watanabe et al.,
2009; Harrison et al., 2018).

On 6 August 2017, a flood event related to glacier lake outburst affected the Zackenberg River (NE Greenland), leaving behind
serious geomorphological impacts on the riverbanks and channel morphology (see Tomczyk et al., 2020). Here, we provided
a very high-resolution dataset representing time-series of data captured immediately before (5 August 2017), during (6 August
2017), and after (8 August 2017) the flood. This unique set of data makes it possible to study the immediate landscape response
to the GLOF event and can be used as a baseline for any long-term monitoring exercise. Our dataset covers approximately a
2.1-km-long distal section of the Zackenberg River. Available files comprise: (1) unprocessed images captured using an
unmanned aerial vehicle (UAV): https://doi.org/10.5281/zenodo.4495282 (Tomczyk and Ewertowski, 2021a); and (2) results
of structure-from-motion (SfM) processing (orthomosaics, digital elevation models, and hillshade models in a raster format),
uncertainty    assessments    (precision    maps)    and    effects    of    geomorphological    mapping    in    vector    format:
https://doi.org/10.5281/zenodo.4498296 (Tomczyk and Ewertowski, 2021b). The availability of unprocessed images means
that the potential user can derive their own photogrammetric products using more advanced technologies (potentially available
in the future) to ensure coherence with future-collected monitoring data.

Potential applications of the presented dataset include: (1) assessment and quantification of landscape changes as an immediate
result of glacier lake outburst flood (Tomczyk and Ewertowski, 2020; Tomczyk et al., 2020); (2) long-term monitoring of

high-Arctic river valley development (in conjunction with other datasets); (3) establishing a baseline for quantification of geomorphological impacts of future glacier lake outburst floods; (4) assessment of geohazards related to bank erosion and debris flow development (hazards for research station infrastructure – station buildings and bridge); (5) monitoring of permafrost degradation; (6) modelling flood impacts on river ecosystem, transport capacity, and channel stability.

## 2 Data acquisition

### 2.1 Study area

The Zackenberg River is located in northeast Greenland (74°30' N; 20°30' W) (Fig. 1a, b). The river is approximately 36 km long, and its catchment covers 514 km$^2$, 20% of which is glaciated. Water sources include melting glaciers, snowmelt, thawing of permafrost, and precipitation (Søndergaard et al., 2015; Kroon et al., 2017; Christensen et al., 2020). One of the Zackenberg River's characteristics is regular floods during summer related to sudden lake drainage — probably due to rupture of the glacier dam (see Jensen et al., 2013; Behm et al., 2017). Between 1996 and 2018, 14 extreme flood events with discharges of over 100 m$^3$ s$^{-1}$ were recorded (Kroon et al., 2017; Tomczyk and Ewertowski, 2020), while two additional ones were observed in the winter period (Kroon et al., 2017). Such events had an enormous impact on the riverscape geomorphology (Tomczyk and Ewertowski, 2020; Tomczyk et al., 2020), discharge and sediment transport (Hasholt et al., 2008; Søndergaard et al., 2015), and delivery of nutrients and sediments into the fiords and delta development (Bendixen et al., 2017; Kroon et al., 2017). In this context, the given dataset aims to establish a baseline for monitoring the consequences of future extreme floods by documenting the state of the riverscape before, during, and after the 2017 glacier lake outburst flood.

### 2.2 UAV surveys

According to the guidelines for using structure-from-motion (SfM) photogrammetry in geomorphological research (see James et al., 2019), details about UAV surveys are presented in Sect. 2.2, and the parameters used for SfM processing are detailed in Sect. 3. In that way, other researchers can use the data to replicate our results; alternatively, as new approaches become available, novel processing methods can be utilised.

**Figure 1: Location of the study area (Reprinted from Journal of Hydrology, Vol 591, Tomczyk et al., Geomorphological impacts of a glacier lake outburst flood in the high arctic Zackenberg River, NE Greenland, 125300, Copyright (2020), with permission from Elsevier).**




### 2.2.1 Rationale


There were three primary goals for conducting the UAV surveys: (1) collect data that would enable quantifying medium-term changes in the river landscape – compared to the available high-resolution 2014 data (COWI, 2015); (2) to document river state and immediate landscape response during the 2017 flood; and (3) to establish a baseline for the monitoring of geomorphological changes in response to future glacier lake outburst floods, including potential geohazards to research

infrastructure (i.e. bridge and buildings of the research station). To achieve these aims, it was necessary to collect data with high spatial resolution, preferably better than 0.05 m ground sampling distance (GSD) (Fig. 2), covering a relatively long section (2.1 km) of the river from the bridge to the delta. We decided to use a small portable UAV as it was more economical in terms of financial and time requirements compared to terrestrial laser scanning (TSL).

### 2.2.2 Equipment

We used a lightweight, consumer-grade UAV – multirotor DJI Phantom 4 Pro. The little weight (1.4 kg) combined with a small size (0.35 m diagonal) ensure that the UAV could be easily transported in the field using a backpack – this was essential, as mechanised transport is not allowed due to fragility of the vegetation. The UAV was equipped with a 20MP, 1-inch size RGB sensor and a global shutter (Table 1). There was a prime lens with 8.8 mm focal length (24 mm equivalent for 35 mm), aperture range from f/2.8 to f/11 and autofocus. A 3-axis (pitch, roll, yaw) gimbal stabilised the camera, enabling it to take

sharp pictures while the craft was in motion. The UAV was equipped with a global navigation satellite system (GNSS) receiver, capable of receiving signals from GPS and GLONASS satellite positioning systems.

### 2.2.3 Survey design and execution

To collect the necessary data, we designed an initial survey plan comprised of five lines approximately parallel to the main river channel's course routed over the centre of the main channel and both banks. During the surveys, this design was modified,

as the river sections containing meandering segments were too wide to be captured with five lines of images with necessary overlap. Therefore, we turned to surveying N-S lines of the images, covering both the river channel and its neighbourhood.

**Figure 2: Comparison of different data sources and their potential for mapping geomorphological features: (a) hillshade model 1 m GSD; (b) hillshade model 0.5 m GSD; (c) hillshade model 0.04 m GSD (generated from UAV-captured images); (d) Planet satellite imagery 3 m GSD (Planet Team, 2017); (e) high-resolution satellite image 0.5 m GSD (© Google Maps 2021); (f) orthomosaic 0.02 m GSD (generated from UAV-captured images)**




Individual flights were operated manually, using DJI GO 4 app for Android, for in such high latitudes onboard GNSS and magnetometer were potentially prone to erroneous reading. Related unexpected behaviours (e.g., errors in compass reading or

lost GNSS signal) were easier to tackle in the manual than automated mode. We captured mostly nadir images with a high overlap (> 80%). Additional oblique images were collected to cover the steep, near-vertical riverbank sections so as to ensure their proper representation in the model. Due to the length of the studied river section, and to comply with the visual line of sight (VLOS) flight operations, three take-off/landing sites were used each day. The weather condition for each day was good, and illumination conditions were sunny. The UAV surveys were performed at average nominal altitudes (from 70 to 110 m

above ground level) to achieve the desired GSD (Table 1). In total, 1972 images were taken on 5 August 2017 (before-flood dataset), 887 images on 6 August 2017 (during-flood dataset), and 1929 images on 8 August 2017 (after-flood dataset). As the river level was fluctuating during the flood (6 August survey), we used a higher flight altitude, which translated into a lower number of images captured on 6 August, but enabled us to cover the area more quickly with approximately the same water level during the survey. So, it was a compromise between photogrammetric quality (i.e., the image network geometry), desired

GSD, and rapidly changing flood conditions.

The unprocessed images captured during the surveys are available at: https://doi.org/10.5281/zenodo.4495282 (Tomczyk and Ewertowski, 2021a). They can be used by interested parties to generate their own photogrammetric products using different methods and/or software than those described in Sect. 3.

**3 Data processing**

**3.1 Structure-from-Motion processing**

The UAV-captured images were processed using Agisoft Metashape Professional Edition 1.5.2. The values used for processing settings in each step were the following:

1) Camera settings – camera type: Frame; enable rolling shutter compensation: unchecked (as the UAV was equipped
140        with global shutter)

2) Images alignment and sparse point cloud generation – accuracy: High; generic preselection: Yes; reference preselection: Yes; key point limit: 100,000; tie point limit: 0 (i.e., unlimited)

3) Gradual selection and removal of the outliers and erroneous points – three-stage selection based on reconstruction uncertainty: 10; reprojection error: 0.5; projection accuracy: 6

4) Optimisation of the sparse point cloud – parameters: f, b1, b2, cx, cy, k1, k2, p1, p2

5) Dense point cloud generation – quality: High, Depth filtering: Aggressive

6) DEM generation – source data: Dense Cloud, Interpolation: Enabled

As we were not able to collect high-quality ground control points (we did not have access to cm-accuracy survey equipment, and it was not possible to cross the river during the flood), control points (CPs) were then generated post-survey using previous





UAV dataset from 2014 (COWI, 2015). In total, 100 points were selected, located mostly on stable, flat boulders, which were easy to identify in the images. Distribution of CPs was along both sides of the river to ensure that the distance between individual points is less than 100 m, which was suggested as optimal by Tonkin and Midgley (2016). The projection used was UTM 27N. The number of points used as "control" to optimise the exterior orientation was: 61 (5 August), 57 (6 August), 61 (8 August). The remaining points were used as independent checkpoints: 39 (5 August), 21 (6 August), 22 (8 August). A

smaller number of points used for data collected on 6 August and 8 August were related to differences in coverage.

### 3.2 SfM processing results

The produced tie points clouds consisted of between 1.2 million (6 and 8 August) and 1.4 million (5 August) filtered points, with low tie point reprojection errors from 0.28 to 0.44, which was indicative of the high quality of the image geometry network (Table 1). Dense cloud point density varied from 322 points $m^{-2}$ (6 August) to 778 points $m^{-2}$. These translated to orthomosaics

with GSDs from 0.018 m (5 August) to 0.028 m (6 August), and DEMs with GSDs from 0.036 m to 0.056 m (Fig. 3). RMS discrepancies for control points and checkpoints were between 0.12 m and 0.15 m, which was expected, as the control and checkpoints were transferred from previously existing data. The coherence between models was also estimated based on test areas selected in stable fragments of moraine and palaeo-delta to ensure significant systematic differences in elevation between datasets do not exist. The final products of SfM processing (Orthomosaic and DEMs) and their derivative (hillshade models)

for each data are available at: https://doi.org/10.5281/zenodo.4498296 (Tomczyk and Ewertowski, 2021b).

**Table 1: Outline of UAV surveys' parameters, processing errors and final products' characteristics following the guidelines suggested by James et al. (2019).**

|  | Survey date | | |
|  | 05 AUG | 06 AUG | 08 AUG |
| --- | --- | --- | --- |
| Camera model | | FC6310 | |
| Sensor size (mm) | | 13.2 x 4.62 | |
| Image size (pixels) | | 5464 x 3640 | |
| Focal length (mm): nominal (35 mm equivalent) | | 8.8 (24) | |
| Pixel size (μm) | | 2.42 | |
| Camera shutter type | | Mechanical, global | |
| coverage (km²) | 0.97 | 1.18 | 0.96 |
| Average flight height above ground level (m) | 71 | 109 | 87 |
| No. of images | 1972 | 887 | 1929 |



| Ground sampling distance (cm pix$^{-1}$) | 1.79 | 2.78 | 2.21 |
|---|---|---|---|
| Number tie points after filtration | 1,438,453 | 1,158,310 | 1,173,564 |
| Tie point RMS reprojection error (pix) | 0.29 | 0.44 | 0.28 |
| Average tie point multiplicity | 4.57 | 4.90 | 4.76 |
| Mean key point size (pix) | 2.61 | 3.05 | 2.58 |
| Dense cloud points density (points m$^{-2}$) | 778 | 322 | 512 |
| No. of control points | 61 | 57 | 61 |
| No. of checkpoints | 39 | 21 | 22 |
| Total (3D) RMSE (cm) on control points | 13.88 | 12.04 | 10.77 |
| Total (3D) RMSE (cm) on check points | 15.33 | 12.16 | 13.30 |
| SD of total (3D) errors (cm) on check points | 6.94 | 4.43 | 5.04 |
| Mean point coordinate precision (mm) [std. dev.]: | | | |
| X | 3.8 [1.5] | 6.1 [3.1] | 4.3 [1.8] |
| Y | 3.7 [1.4] | 5.6 [2.99] | 3.9 [1.5] |
| Z | 10.7 [4.3] | 15.3 [7.9] | 11.9 [4.4] |

### 3.3 Mapping

The mapping process was based on the approach proposed by Chandler et al. (2018), i.e. identification and interpretation of the geomorphological features were based on a combined analysis of remote sensing products and their derivatives (orthomosaics, DEMs, slope maps, hillshade models) as well as ground-based truthing. Final vector datasets were vectorised on-screen in ArcMap 10.6 software. The main geomorphological units (e.g., relict fluvial terraces, modern floodplain, slopes) and areas affected by mass movements of various types (e.g., debris flows, debris slumps) were mapped as polygons. Additional layers of polylines included features such as scarps or thermal contraction cracks. River extent (i.e., area covered by water) is provided for each day as a separate polygon layer. Geomorphological features are provided as a separate file for before-the-flood (5 August 2017) and after-the-flood (8 August 2017) dataset. The mapping results in the form of vector files in the shp format (compatible with most GIS software) are available to download from at: https://doi.org/10.5281/zenodo.4498296 (Tomczyk and Ewertowski, 2021b). Vector data combined with the hillshade models were presented as a series of geomorphological maps (see Tomczyk and Ewertowski, 2020 for details)



**Figure 3: Examples of delivered dataset illustrating before and after the flood situation: (a, e) digital elevation model; (b, f) hillshade model; (c, g) orthomosaics; (d, h) results of geomorphological mapping**






## 4 Quality assessment and known limitations

Quality assessment based on data presented in Table 1 indicates a high quality of internal image network geometry, illustrated by low sub-pixel values of tie points reprojection errors. Room-mean-square-errors (RMSE) and standard deviations (SD) of errors on checkpoints are generally between 0.12 and 0.15 m. Although such values are higher than the GSD of all datasets

(between 0.018 and 0.028 m), such magnitude of errors was considered acceptable for the quantification and mapping of landscape changes, especially as between 5 August and 8 August the resultant lateral erosion of riverbanks from the flood reached almost 10 m in some sections, (see Tomczyk et al., 2020 for details). If necessary, lower values of errors can be achieved in the future if additional ground control points are surveyed using a cm-accuracy survey equipment.

To estimate the spatial variability of the models' photogrammetric and georeferencing uncertainties, the precision estimates for sparse point clouds were generated in Agisoft Metashap and exported using the Python script provided by James et al. (2020). The precision analysis indicated that the vertical component was less spatially consistent than the horizontal ones for all three surveys (Fig. 4). For the models' ground parts, the overall precision was limited by the precision of control points, which is not surprising as they were derived from the older, less detailed remote sensing dataset. The mean point precision

estimates varied from 4 to 6 mm for the horizontal component and from 11 to 15 mm for the vertical one (Table 1) – the weakest values were for the 6 August 2017 dataset, which was expected as the average flying altitude was highest then. Precision maps are available to download from at: https://doi.org/10.5281/zenodo.4498296 (Tomczyk and Ewertowski, 2021b). Z discrepancies on control points were calculated using Doming Analysis software (v.1.0) (James et al., 2020). The analysis indicated no doming distortion (Fig. 5), which is probably related to the generally very high overlap of images and

the inclusion of oblique images of the steep riverbanks.



**Figure 4: Precision estimates for X, Y, Z coordinates of tie points**




**Figure 5: Spatial distribution of errors on control and check points: (a) Z-error against radial distance from the tie point centroid; (b) Z-error by colour in plan view (X, Y are distanced from tie points centroid).**



Individual orthomosaics and DEMs were also inspected, resulting in the discovery of the following several problems, which
ought to be taken into account in any future analysis:

1) In general, the interpretation of riverbank conditions can be tampered by vegetation cover and/or bank undercutting (Niedzielski et al., 2016; Hemmelder et al., 2018). The proposed solutions included taking the mean elevation value of the bank in between the vegetated areas and then using it as the reference height (Hemmelder et al., 2018) or interpolating a line between the last exposed sections of the riverbank, not covered by trees and bushes (Niedzielski
et al., 2016). While vegetation cover is usually not a problem in the case of Arctic rivers, such an approach might be useful when other obstacles (e.g., shadows, infrastructure) prevent the direct measurements of the bank's heights. In the case of the presented dataset, some sections of riverbanks were steep, near-vertical, before the flood. However, during the flood, some of the sections were significantly undercut, forming deeply incised niches – these overhanging banks obstructed the view of the bottom part of some studied sections from the air. During the UAV campaigns, we
took oblique images to at least produce a proper representation of steep slopes; however, it was not possible to take horizontal images due to the presence of water. As a result, it was impossible to calculate the volume of sediments eroded from the niches under these overhanging sections.

2) Structure-from-motion is based on reconstructing the image network geometry based on characteristic points that appear in several images (Westoby et al., 2012). It therefore fails where there are rapidly moving objects, which
changed their position in time between the images captured. The structure-from-motion photogrammetry can reconstruct the location of points in dry areas, and, in the case of transparent water, also points located underwater (Carrivick and Smith, 2019). However, in our study, the high turbidity of water and sediment suspension prevented viewing of the riverbed. As an Arctic river, the Zackenberg River has suspended sediment concentrations within a range of 50 to 500 mg $L^{-1}$ (Søndergaard et al., 2015), which can increase even up to 4000 mg $L^{-1}$ during glacial lake
outburst floods, indicated by the lack of transparency and the yellowish or brownish colours of water in the orthomosaics. The turbidity of water is also very high (Ladegaard-Pedersen et al., 2017), as was also found in our surveys. The fact that the water surface was full of ripples gave rise to bi-directional reflectance problems. Therefore, it was not possible to adequately resolve the surface of flowing water. To partly address this issue, the water bodies were masked from DEMs and hillshade models. They are, however, visible in orthomosaics which enables the user
to assess the character of water flow.

3) Some fragments of the models revealed artefacts associated with mismatches in points generation. These areas can generate erroneous elevation values, which can be identified in the DEM and hillshade model as unexpectedly rough surfaces in places where the ground level should be uniform. These areas were indicated with polygon files for easy identification in case of future analysis.


**Figure 6: Examples of encountered problems: (a) undercut/overhanging river sections; (b) rapidly moving water; (c) artefacts related to errors in surface reconstruction.**



## 5 Data and code availability

All described data are available in the Zenodo repository. The structure of the dataset is as follows:

1)   Unprocessed UAV-captured images (~46 GB) are available at: https://doi.org/10.5281/zenodo.4495282 (Tomczyk and Ewertowski, 2021a). The images are zipped into three folders following naming convention: 2017_08_05_before_flood_unprocessed_UAV_images, 2017_08_06_during_flood_unprocessed_UAV_images, 2017_08_08_after_flood_unprocessed_UAV_images. The images are in jpg format and contain embedded positions in geographic coordinate system WGS84 obtained from the on-board GNSS receiver.

2)   The results of photogrammetric processing (~18 GB) are available at https://doi.org/10.5281/zenodo.4498296 (Tomczyk and Ewertowski, 2021b) in the file Sfm_products.zip, and are grouped into subfolders with the following names: dem (containing digital elevation models), orthomosaic (containing orthomosaics), hs (containing hillshade models); all data are in GeoTIFF format in the UTM 27N projected coordinate system. Individual files are named as follows: yyyy_mm_dd_[filetype]_[status].tif, where:

260         a.   yyyy_mm_dd – is a date, e.g. 2017_08_05

        b.   [filetype] – with three values: dem (= DEM), ortho (= orthomosaic), hs (= hillshade)

        c.   [status] – with three values: before_flood, during_flood, after_flood

    3)   The mapping results (25 MB) are in the same repository entry as SfM processing results, i.e. at https://doi.org/10.5281/zenodo.4498296 (Tomczyk and Ewertowski, 2021b) in the folder "mapping.zip". Inside, there

are four sub-folders:

        a.   General – contains general vectors that did not change over the course of three days (e.g., station buildings, 4x4 trail)

        b.   River_extent – contains polygons for river extent for 2014 (generated from older UAV data (COWI, 2015)), and for 05, 06, and 08 August 2017. The 2017 data are named as yyyy_mm_dd_river

270         c.   Before_flood_geomorphology – contains polygon and lines illustrating geomorphological features before the flood with separate files providing extent of mass movements which can be potentially hazardous e.g., debris flows, debris falls, rockfalls, slumps (names of individual files are provided in Table 2)

        d.   After_flood_geomorphology – contains polygon and lines illustrating geomorphological features after the flood with separate files providing extent of mass movements which can be potentially hazardous e.g., debris

275          flows, debris falls, rockfalls, slumps (names of individual files are provided in Table 2)

     All data are in shp vector format in the UTM 27N projected coordinate system.

    4)   Precision estimates for tie points and precision maps for X, Y, Z coordinates are in the same repository entry as SfM processing results, i.e. at https://doi.org/10.5281/zenodo.4498296 (Tomczyk and Ewertowski, 2021a) in the folder " uncertainty_assessment.zip". Individual files are named as follows:



    a.   yyyy_mm_dd_[before_flood/during_flood/after_flood]_points_precision.txt – files contain precision estimations for each tie point

    b.   yyyy_mm_dd_[before_flood/during_flood/after_flood]_[X/Y/Z]_precision.tif – files contain precision for each coordinate as raster file

Structure-from-motion processing was performed in the proprietary software Agisoft Metashape (https://www.agisoft.com/). Mapping was performed in ArcMap (https://www.esri.com/en-us/arcgis/about-arcgis/overview). Python script exporting presion estimates from Agisoft Metashape and Doming Analysis software (v.1.0) (James et al., 2020) are available to download from https://www.lancaster.ac.uk/staff/jamesm/software/sfm_georef.htm.

**Table 2: List of filenames for corresponding dates and content**

| Filename | | Content description |
|---|---|---|
| **General files folder:** | | |
| 2017_4x4_track.shp | | Track accessible to station vehicle |
| 2017_bridge.shp | | Location of the pedestrian bridge across the Zackenberg River |
| 2017_thermal_contraction_cracks.shp | | Thermal-contraction cracks |
| **River_extent folder:** | | |
| 2010_river_mask.shp | | Extent of the river vectorized from 2014 data (COWI, 2015) |
| 2017_08_05_before_flood_land_mask.shp | | Extent of the land area in before-flood orthomosaic |
| 2017_08_05_before_flood_river_mask.shp | | Area covered by water in during-flood orthomosaic |
| 2017_08_06_during_flood_land_mask.shp | | Extent of the land area in during-flood orthomosaic |
| 2017_08_06_during_flood_river_mask.shp | | Area covered by water in during-flood orthomosaic |
| 2017_08_08_after_flood_land_mask.shp | | Extent of the land area in after-flood orthomosaic |
| 2017_08_08_after_flood_river_mask.shp | | Area covered by water in after-flood orthomosaic |
| **Geomorphological features** | | |
| 05 August 2017 (before flood) | 08 August 2018 (after flood) | |
| 2017_08_05_before_flood_ mass_movement_lines.shp | 2017_08_08_after_flood_ mass_movement_lines.shp | Linear elements of mass-movement-related features (active fluvial scarps, stable fluvial scarps, old failure scarp) |
| 2017_08_05_before_flood_ mm_debris_fall.shp | 2017_08_08_before_flood_ mm_debris_fall.shp | Landform related to debris fall activity |
| 2017_08_05_before_flood_ mm_debris_flow.shp | 2017_08_08_before_flood_ mm_debris_flow.shp | Landform related to debris flow activity |



| - | 2017_08_08_before_flood_ mm_rockfall.shp | Landform related to debris rockfall activity |
|---|---|---|
| 2017_08_05_before_flood_ mm_slump.shp | 2017_08_08_before_flood_ mm_slump.shp | Landform related to debris slump activity |
| 2017_08_05_before_flood_ morphology_polygons.shp | 2017_08_08_after_flood_ morphology_polygons.shp | Morphological units stored as polygons (e.g. modern floodplain, alluvial fan, relict fluvial terrace, flat area, gentle bank, steep bank) |
| 2017_08_05_before_flood_ surface_runoff_traces.shp | 2017_08_08_after_flood_ surface_runoff_traces.shp | Traces of surface runoff |

## 6 Conclusions

The ability to detect changes in the geomorphology of the riverbed and riparian areas remains a crucial issue in monitoring and modelling the geomorphic effects of flood events. Using a UAV survey for rapid assessment, as in the case of the studied 2017 flood, can be more beneficial than other methods (like high-resolution satellite imagery, terrestrial laser scanning) (cf.
Carrivick et al., 2016; Smith et al., 2016), as it allows for covering the substantial length of the river with high-resolution data. Such data are intended to be a baseline for future monitoring projects. Potential applications of the presented dataset include:

1) Establishing a long-term monitoring of high-Arctic river valley development in a permafrost terrain – climate warming in the Arctic is more intense than in other regions (see Moritz et al., 2002; Walsh et al., 2011; Duarte et al., 2012), with the thawing of permafrost in Greenland being one of the effects (Elberling et al., 2013; Anderson et al.,
2017). In such a dynamic environment, riverscapes are also likely to transform rapidly (Chassiot et al., 2020). As our data covers river section located close to the Zackenberg Research Station, it facilitates logistics and can potentially enable developing long-term remote sensing data series illustrating the dynamic response of the riverscape to ongoing climate change, which is essential from the standpoint of long-term landscape evolution.

2) Quantification, monitoring and modelling of geomorphological impacts of glacier lake outburst flood – the presented
dataset was meant to quantify changes related to the 2017 GLOF (see Tomczyk and Ewertowski, 2020; Tomczyk et al., 2020); however, these studies only described the immediate impacts of a single flood event. Using the provided dataset as a baseline for the monitoring of future changes, it should be possible to quantify the difference between geomorphological effects of "normal" (i.e., high-frequency, low-magnitude) processes on the one hand, and extreme (i.e., low-frequency, high-magnitude) events on the other. Also, by linking the intensity of a geomorphological
response to hydrological data about flood characteristics, it should be possible to improve modelling routines (cf. Carrivick, 2007a, b; Carrivick et al., 2011; Guan et al., 2015; Staines and Carrivick, 2015).

3) Geo-hazards assessment – the Zackenberg Research Station premises are located close to the riverbank which is regularly affected by floods. The development of debris flows, which has started to threaten the Station's



infrastructure, is one outcome of the removal of sediments from the channel by flood. Another example of geohazards
is washing out the foundation of the bridge located up the valley. These hazards require regular monitoring to prevent
damage to the infrastructure, and the presented database can be used to assess current hazards and establish a baseline
for future monitoring.

**Author contributions**

AMT and MWE collected data during field campaign and performed the photogrammetric processing and uncertainty analysis.
AMT mapped the geomorphology and wrote the paper with inputs from MWE.

**Competing interests**

The authors declare that they have no conflict of interest.

**Acknowledgements**

We are very grateful for the support from INTERACT Network, which allowed us to visit Zackenberg Research Station in
2017 - The research leading to these results has received funding from the European Union's Horizon 2020 project INTERACT,
under grant agreement No. 730938, project number: 119 [ArcticFan]. The realisation of the fieldwork would not have been
possible without logistic support provided by the crew of the Zackenberg Research Station.

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
