# Peer review of "Baseline data for monitoring geomorphological effects of glacier lake outburst flood: A very high-resolution image and GIS datasets of the distal part of the Zackenberg River, northeast Greenland"

_Earth System Science Data, 2021_

## Referee Comment (RC4)

Review Tomczyk and Ewertowski

This is a data description of a detailed dataset regarding a GLOF event in Zackenberg. Albeit surely unique, I am wondering if the accuracies hold in an absolute sense, as no GCPs are taken and creep may have been significant and unpredictable since 2014. If this doesn't hold, it is also not a dataset which will be useful for many follow-up studies.

In that respect it should warrant publication as a one-process dataset. There, however, I see too many papers using it already, of which 2 are Journal papers, and 2 Zenodo repositories.

Tomczyk, A. M., and Ewertowski, M. W.: UAV-based remote sensing of immediate changes in geomorphology following a glacial lake outburst flood at the Zackenberg river, northeast Greenland, J Maps, 16, 86-100, doi:10.1080/17445647.2020.1749146, 2020.

Tomczyk, A. M., Ewertowski, M. W., and Carrivick, J. L.: Geomorphological impacts of a glacier lake outburst flood in the high arctic Zackenberg River, NE Greenland, J Hydrol, 591, 125300, doi:10.1016/j.jhydrol.2020.125300, 2020.

Tomczyk, A. M., and Ewertowski, M. W.: Before-, during-, and after-flood UAV-generated images of the distal part of Zackenberg river, northeast Greenland (August 2017), doi:10.5281/zenodo.4495282, 2021a.

Tomczyk, A. M., and Ewertowski, M. W.: Before-, during-, and after-flood UAV-generated digital elevation models, orthomosaics, and GIS datasets of the distal part of Zackenberg river, northeast Greenland (August 2017) doi:10.5281/zenodo.4498296, 2021b.

I really don't see a reason why to recycle this once more, particularly, since I am doubtful that it really gives the absolute accuracy required for follow-up studies.

Figures are graded colourmaps and I don't think they qualify for barrier-free colour codes for colour blinds. Also, they don't show categories and it is hard to make use of them.

L8 2x Arctic. Is ‚intense' good wording here?

L9: sounds as if climate warming was neither low-freq nor high-magn. Suggest expanding

L10: grammar: singular/plural!

L18: of a glacier lake…

Intro

L25: riverscape evolution… sounds weird to me

L33: and commonly occur: something with grammer there

L34-36: too many refs that refer to a very broad statement

L38: of a moraine dam

L44: in the case of Zack, I believe we will rather see the opposite with glacier thinning??

L48: related to a glacier lake…

L48: leaving behind serious… sounds jargon to me

L61-66: direct repetition of abstract.

L71: suggest 'glacier-covered' instead of glaciated

L69: ZR could warrant an abbreviation

L74: check refs and GEM database. It is >200 to my knowledge.

L77: do Bendixen et al refer to ZR specifically? Check also Landegaard-Pedersen 2017 for the sediment part

Fig 1: Is it Young Sund or Young Sound in English? Unsure. What is the basis for the isolines? Acquisition date of ice margin extent? The bridge is referred to in the text but not in the map

L91: what is medium-term in this context?

L91-95: repetition to the above?

L97: delete: relatively long?

L98-99: delete. It comes later. No ref to TLS (wrong abbreviation used) necessary in my opinion if you don't use it. If at all in the discussion

L100-106: add brand of sensors

Fig. 2: show in Fig 1 extent. Is it really 0.5 m you get from GoogleMaps? Doublecheck

L123: what are good weather conditions? Specify or omit

L129: remove: 'so'

L148: why? It is part of the station infrastructure. In 2017 the bridge was there, was it not possible to cross? I am surprised and a bit doubtful by the lack of GCPs

L150: This is all permafrost, even large boulders may move due to creep, definitely between years. I am doubtful that the accuracy can be achieved down to say 4 mm (Tab1) if no GCPs are taken. Even if we refer to the 0.12 m to 0.15 m this may be within the moving conditions of the terrain over 4 summer seasons if you use the COWI DEM. This warrants at least discussion.

L158: which units?

L173: ground-truthing

L173: vector were vectorized... reword

Fig 3: the color scale of a and e is not useful: use contours and descrete colours as well as contour labels. Also very strange: after the flood the elevation of the orographically left river bank is in the order of 13 m higher than the right river bank?? I have some doubts that this is realistic undercut, also when I look at 3g. if that was the cassse, the yellow colours would likely disappear in 3e and just the plateau remains. Mark extent in fig 1. Scarp and drainage is strange to me here. What is the water body in the upper left corner of the fig? Does a map with different colours related to the slope qualify as 'geomorphological mapping'?

L189: how do you relate this accuracy to potential permafrost creep?

L194: by then they will have moved again to some extent I fear. It is necessary in such an environment to have DGPS GCPs for each survey if such high accuracy should be obtained.

L195-205: please make it very clear what is precision within the 3 DEMs and what is the absolute precision.

Fig 4: the colour scale is unacceptable. Have contours or discrete ones. I cant see anything here. Also indicate where you are in an overview fig

Fig 5: same as above. Unclear what is shown. which tie point centroid? So here we have errors of up to 0.4 m?

L216-220: it is really unclear why this is relevant there. Stick to topics that occur

L224: do you have pictures of this undercut? It is a bit hard to believe to me

L235: yellowish is jargon I believe

L240: how can you assess the character of water flow from orthophotos??

L287: precision? Typo

Tab 2: why not showing the vectors in all maps (bridge, track). Is it really a 4x4 track??

L295-317: largely another repetition

---

## Author Response (AR1)

**Dear Editor,**

**Dear Referees,**

We would like to thank you for your constructive comments and remarks. We answered reviewers' comments and modified the manuscript according to the suggestions:

1) As proposed by Referee #1, we added information about the frequency of GLOFs at Zackenberg.

2) Following the suggestions of Referee #2, we provided a brief paragraph about GLOFs in Greenland, added information about summer discharges at Zackenberg, and added a figure (Fig. 9) with an example of change detection analysis based on debris flows located close to the research station.

3) The biggest concern of Referee #3 was the limited number of potential data users. We explained and demonstrated that our dataset is not only interesting for researchers studying the Zackenberg river but also for scientists interested in broader process-based research on flood effects. Regular GLOFs at Zackenberg make this area an ideal field-based "experiment" site to study processes such as sediment transport, entrainment, and erosion and also impact of floods on ecosystems and delivery of organic matter from permafrost - and our dataset can be used to establish of long-term monitoring. Zackenberg Research Station's proximity can facilitate this long-term monitoring by logistics support and access to the Arctic environment, which is otherwise relatively hard to access and survey. Moreover, Zackenberg Research Station hosts many researchers every year interested in utilizing the presented dataset as a background and for further in-depth analysis. Therefore, considering researchers visiting Zackenberg Station and scientists interested in broader process-based research, our dataset will attract a sufficient number of users to fulfil ESSD goals.

4) Referee #4 concerns were related to the lack of DGPS ground control points. We explained that our processed datasets were registered using coordinates from the onboard GNSS system further constrained by control points collected from older surveys. Therefore, all surveys were registered to the same space of geographical coordinates to ensure high (dm-scale) external accuracy and very-high (cm-scale) internal precision. Furthermore, our datasets were processed using industry standards and guidelines to ensure the high quality and reproducibility of processed datasets. Moreover, as demonstrated in several studies (e.g., Feurer and Vinatier, 2018; Cook and Dietze, 2019; de Haas et al., 2021), co-alignment of time-series of surveys using structure-from-motion (SfM) processing can provide better relative accuracy than the classic approach of individual SfM processing of each survey using GCP. Therefore, we also provided unprocessed images that can be easily co-aligned and combined with future surveys. Another concern was related to the fact that our data were already uploaded in 2 Zenodo repositories – however, publishing data in an easy to access repository is required by ESSD before submission of the Data description paper.

Please, find the responses to individual comments below, indicated using the following key:

**Bold – referee's comment**

*Italic – our response*

Normal text – modifications of the manuscript

We believe that our explanations and modifications of the manuscript will convince the Editor and Referees that our data set is of high quality and will attract a sufficient number of end-users to be suitable for publication in ESSD.

Yours Sincerely,

Aleksandra Tomczyk and Marek Ewertowski

**REFEREE #1**

Dear Reviewer,

Thank you very much for your positive comments and review. Please, find the response to your comments below:

1. **Maybe the authors could add some more discussion on the expected frequency (recurrence intervall) of the high-magnitude low-frequency event presented in this data description paper,**

*We added the following information:*

The first GLOF at Zackenberg was observed in 1996 and since then floods occurred every year or at the two-year interval (Kroon et al., 2017; Tomczyk and Ewertowski, 2020). The lake, which is the source of GLOF, is located more than 3 km from the current ice margin, so we expect a similar or higher frequency as more water will be melting from glaciers and stored in the lake. Thus, future monitoring is needed to investigate whether the GLOFs will be observed more frequently but with lower discharge magnitude or less often but with higher discharge.

2. **and on the quantitative importance of this high-magnitude low-frequency event as compared to low-magnitude high-frequency events in the investigated study site.**

*We added the following paragraph:*

As the high-magnitude low-frequency events are typically rare and difficult to predict, our understanding of the quantitative aspect of geomorphological changes related to them remains limited compared to the "normal" processes (Tamminga et al., 2015b). These arise particularly from difficulties in collecting high-resolution data before and after these innately unpredictable and rare flood events. However, investigation into the geomorphological response of river morphology to "extreme" events is key to understanding the evolution of river morphology and being crucial from the standpoint of river modelling and monitoring (Tamminga et al., 2015a; Tamminga et al., 2015b). Moreover, the relationship between the magnitude of the flood and geomorphological effects is not fully understood. For example, in the case of Zackenberg River, immediate (2-days) lateral erosion compared to three-year erosion was spatially very diversified. In some sections, immediate lateral erosion after the 2017 flood reached up to 10 m, whereas the same section was stable between 2014 and 2017, even though higher peak discharges characterised 2015 and 2016 GLOFs than 2017 GLOF (Tomczyk et al., 2020). Further process-based studies are necessary to observe and model links between the magnitude of a flood and the severity of erosion. It is especially important in periglacial landscapes where lateral bank erosion can be responsible for delivering a large quantity of organic matter and widespread changes in ecosystems. especially combined with other weather extreme events (see Christensen et al., 2021)
* * *
**REFEREE #2**

Dear Professor Petrakov,

Thank you very much for your positive and constructive review. Please, find the responses to detailed comments below:

3. **In the Introduction it will be better to focus a bit more on glacier-related floods in the Arctic, especially in the Greenland, provide some more details on their observed and projected frequency and magnitude as well as on their geomorphic effect.**

*We added the following paragraph to the Introduction:*

"GLOFs in Greenland were reported from several locations (see Carrivick and Tweed, 2019 for more detailed review), including Lake Isvand (Weidick and Citterio, 2011), Russel Glacier (e.g., Russell, 2009; Russell et al., 2011; Carrivick et al., 2013; Carrivick et al., 2017; Hasholt et al., 2018), Kuannersuit Glacier (Yde et al., 2019), Lake Tininnilik (Furuya and Wahr, 2005), Lake Hullet (Dawson, 1983), Qorlortorssup Tasia (Mayer and Schuler, 2005), Zackenberg river (Søndergaard et al., 2015; Kroon et al., 2017; Ladegaard-Pedersen et al., 2017), Catalina Lake (Grinsted et al., 2017). Estimated water volume losses varied from ~5 x $10^6$ $m^3$ to ~6400 x 10 $m^3$, while peak discharges could reach up to ~1430 $m^3s^{-1}$ (Dawson, 1983; Furuya and Wahr, 2005; Russell et al., 2011; Carrivick et al., 2013; Søndergaard et al., 2015; Carrivick and Tweed, 2019). The frequency of GLOFs in Greenland varies from annual through decades (e.g., Zackenberg River, Russel Glacier, Lake Tininnilik) to one-time events (e.g., Kuannersuit Glacier) (Furuya and Wahr, 2005; Russell et al., 2011; Carrivick and Tweed, 2019; Yde et al., 2019). The most significant geomorphological and hydrological effects included the formation of bedrock canyons and spillways, transport of large boulders, riverbanks erosion, development of coarse-sediment bars and deltas, outwash surfaces, and ice-walled canyons (Russell, 2009; Carrivick et al., 2013; Carrivick and Tweed, 2019; Yde et al., 2019). Despite numerous reports, so far, no detailed topographical data of a river system exists, which could serve as a baseline for long-term monitoring of landscape changes to understand, quantify and model changes resulting from GLOF in comparison to normal-frequency processes. "

4. **Lines 73-76 – it is not clear what discharge is normal during summer time. Some more details on river regime should be provided.**

*We added information about summer discharges:*

Typical discharges during summer month were from 20 $m^3$ $s^{-1}$ to 50 $m^3$ $s^{-1}$, and usually lower at the end of melting season (September-October) (Søndergaard et al., 2015; Ladegaard-Pedersen et al., 2017)

5. **Fig.3. It might be better to show before-the-flood river channel on "After flood" series of maps, it makes this figure more reader friendly.**

*Figure modified as suggested*

[Figure]

**Figure 3: Examples of delivered dataset illustrating before and after the flood situation: (a, e) digital elevation model; (b, f) hillshade model; (c, g) orthomosaics; (d, h) results of geomorphological mapping**

6. **It will be great to add one more figure with DEM difference for some erosion and accumulation areas to provide more data on geomorphological effect of the flood as well to add some a brief information on erosion/ entrainment/ accumulation rates depending on channel slope angle and sinuosity, elevation change etc. despite most of the data has been published in (Tomczyk et al., 2020).**

*We added the figure with an example of DEM of Difference for debris flows developed close to the building of the Research Station. We also added a brief text related to observed geomorphological changes.*

[Figure]

**Figure 9: Examples of DEM of Differences demonstrating geomorphic change detection for two debris flows located in the proximity of Zackenberg Research Station.**

"An example of geomorphic change detection is presented in Fig. 9, demonstrating the acceleration of debris flows resulting from sediment entrainment at the base of the river bans by floodwater. Overall, the observed changes were spatially variable – erosion dominated along steep banks as expected; however, understanding of differences in erosion rates between sites requires further studies, which will

consider differences in lithology as well as modelling of water flow to investigate potential erosion forces in relation to channel characteristics."
* * *
**REFEREE #3**

Dear Reviewer,

Thank you very much for your opinion and comments. Please, find our answers below:

7. **Interest of a broader audience is fairly limited in my opinion; presented data are indeed interesting but potential usage as well as utilisation in other than very specialised and geographically narrowly-focused case studies are nebulous to me**

*Our dataset presents very-high resolution data (better than 0.1 m GSD) illustrating landscape characteristics immediately before, during, and after the flood. To our knowledge, there were no similar datasets available related to large flood events in the Arctic. Such data can be utilised in process-based studies to expand general knowledge about flood-related processes, including modelling sediment entrainment and validation of the models, especially in combination with hydrological data available through Greenland Ecosystem Monitoring Programme. Location of the presented dataset is its particular strength as the close proximity of Zackenberg Research Station infrastructure and regular occurrence of GLOF events make it an excellent natural "laboratory" for investigation and modelling studies, which can be then transferred to other regions enabling assessments of geomorphological and hydrological changes. Our datasets can also be used for modelling ecosystems and habitats under warming climate conditions.*

8. **Assessment and quantification of landscape changes (perhaps the most interesting utilisation) have already been analysed and published by the authors (Tomczyk and Ewertowski, 2020; Tomczyk et al., 2020), further reducing potential use of the dataset**

*We already published quantification of immediate (2-day) changes related to the flood, which is clearly indicated in the data description manuscript, and references to these studies are provided. However, we believe that other applications are equally or more interesting (especially modelling process-based studies, geo-hazards assessments, and ecological studies); therefore, we make our datasets publicly available to facilitate their re-use and implementation in future works. In addition, we provided unprocessed images, so they can be used as a baseline for monitoring exercises and be co-aligned with data collected during future surveys to establish unique, long-term monitoring of landscape changes in detailed spatial scale. As mentioned before, the proximity of Zackenberg Research Station, Greenland Ecosystem Monitoring Programme and relatively regular GLOF events in Zackenberg River are greatly beneficial to achieve this aim.*

9. **Methodological approach is technically sound but not novel nor innovative – there is an array of studies focusing on application of UAVs for the production of very high resolution DEMs across the globe (tens of studies adopting this approach published every year, according to the WOS)**

*Yes, we used the methodology which becomes "industry" standard in geographical studies, and as stated in Sections 2 and 3, we followed guidelines presented in several papers (Chandler et al., 2018; James et al., 2019; James et al., 2020) to ensure quality and reproducibility of our data. However, our dataset is unique because it captured detailed topography of the river system immediately before,*

*during, and after the flood. To our knowledge, no similar dataset was available for large flood events in the Arctic.*

10. **Moreover, the ESSD editors in their definitions of goals, practices and recommendations (https://essd.copernicus.org/articles/10/2275/2018/) state that: 'Authors should know that, to ensure that ESSD products enable substantial advances in future research, editors must apply dual criteria in all cases; does the data as submitted demonstrate sufficient quality and will the data product interest a sufficient number of users? Clearly, a small data set collected over a short time at a single location generally does not qualify ... ' To sum up, I'm not convinced that presented data – though interesting – can be of use for others than a limited number of researchers working in this particular area and - unlike the other two reviewers - I don't find this dataset suitable for ESSD (this should be decided by the editors). I'm also concerned about the previous exploitation of these data in another two publications of the authors.**

*As we mentioned in previous responses, there are several applications of our data themselves, including process-based modelling studies, geo-hazards assessments, ecological studies. Moreover, thanks to the availability of both processed data and unprocessed images, our datasets can be easily extended and combined with future surveys, enabling long-term monitoring. Polar regions are especially vulnerable to climate warming and establishing long-term landscape monitoring is critical to understand processes in such a rapidly evolving environment. Proposed monitoring is facilitated by logistic support of Zackenberg Station. Giving the unique geographical location (i.e., regularly flooded Arctic River, but at the same time relatively easy access), extensive Greenland Ecosystem Monitoring programme, and numerous researchers visiting Zackenberg every year, we expect that our dataset will be re-used frequently in the future and will generate interest of the sufficient number of users.*
* * *
**REFEREE #4**

Dear Reviewer,

Thank you very much for your detailed and constructive review. Please, find detailed responses to your comments below:

General comments:

1. **While it is an interesting dataset, I doubt its replicability since it is in a moving terrain and has no DGPS GCPs.**
   *and*
   **This is a data description of a detailed dataset regarding a GLOF event in Zackenberg. Albeit surely unique, I am wondering if the accuracies hold in an absolute sense, as no GCPs are taken and creep may have been significant and unpredictable since 2014. If this doesn't hold, it is also not a dataset which will be useful for many follow-up studies.**

*As demonstrated in several studies (e.g., Cook and Dietze, 2019; de Haas et al., 2021), a time series of UAV data can be successfully registered without ground control points using the co-alignment approach. According to de Haas et al. (2021), such an approach outperforms the classic method (i.e., each survey processed individually with GCPs) regarding relative accuracy and change detection. The absolute accuracy will be low (several-m), but relative accuracy within co-aligned series of surveys will be high (up to <0.1 m). Therefore, we provided original unprocessed images to enable future users*

*to perform co-alignments with newly collected surveys. Furthermore, our data cover stable grounds (e.g., buildings of the research station and fragments of marine terraces), so there will be enough stable points to co-register future surveys. Moreover, de Haas et al. (2021) demonstrated that even in such an unstable environment as debris flow torrent, the co-alignment approach gave relative accuracy of change detection better by a factor 3 than the classical approach with individually processed surveys with GCPs.*

**2. Furthermore it is as far as I understand already published (in 2 journal papers and 2 zenodo repositories) and hence does not warrant an additional publication in my opinion.**

*Earth System Science Data (ESSD) journal publish articles describing research datasets to facilitate their future re-use. According to ESSD guidelines, datasets presented in the paper must be uploaded to a repository that provides permanent digital object identifiers (DOI) and published under a non-restrictive license. Therefore, we uploaded our data in Zenodo as two repositories (due to the large size of the datasets): one includes raw UAV-generated images and the second with processed products (orthomosaics, DEMs, hillshade, vectors) to fulfil the ESSD guidelines.*

*In general, our dataset comprises three surveys (before-, during-, and after-the-flood). As we provided both unprocessed images as well as final products, there are several potential applications of presented datasets, including:*

*(1) assessment and quantification of landscape changes as an immediate result of glacier lake outburst flood;*
*(2) long-term monitoring of high-Arctic river valley development (in conjunction with other datasets);*
*(3) establishing a baseline for quantification of geomorphological impacts of future glacier lake outburst floods;*
*(4) assessment of geohazards related to bank erosion and debris flow development (hazards for research station infrastructure – station buildings and bridge);*
*(5) monitoring of permafrost degradation; and*
*(6) modelling flood impacts on river ecosystem, transport capacity, and channel stability.*

*Moreover, other applications might be possible in the future if new surveys extend the presented time series – to allow that we provided unprocessed images so they can be co-aligned with future surveys (please, see the response to comment #1). We published two papers (Tomczyk and Ewertowski, 2020; Tomczyk et al., 2020) which serve as a proof of concept and demonstrate utilisation of the presented dataset to application 1 (quantification of geomorphological impacts of the flood) - These papers are clearly indicated in the manuscript. However, so far, no publications use the presented datasets in other of the applications mentioned above; therefore, our datasets should be re-used at least several times or more if the long-term monitoring program will be established.*

**3. Figures are graded colourmaps and I don't think they qualify for barrier-free colour codes for colour blinds. Also, they don't show categories and it is hard to make use of them.**

*We provided alternative versions of the figures with contour lines and colour categories. Changes for each figure and the figures themselves are described in responses to "detailed comments" below.*

Detailed comments:

**4. L8 2x Arctic. Is ‚intense' good wording here?**

*We changed "Arctic" to "polar" and "intense" to "widespread". The sentence is now as follow:*

"The polar regions experience widespread transformations, such that efficient methods are needed to monitor and understand Arctic landscape changes"

**5. L9: sounds as if climate warming was neither low-freq nor high-magn. Suggest expanding**

*We meant hydrological and geomorphological events; the text was modified as follow*

"[…] Arctic landscape changes in response to climate warming and low-frequency high-magnitude hydrological and geomorphological events."

**6. L10: grammar: singular/plural!**

*Corrected:*

*"[...]* are glacier lake outburst floods."

**7. L18: of a glacier lake…**

*Corrected*

**8. L25: riverscape evolution… sounds weird to me**

*Changed to "river system":*

Long-term river system evolution is the effect of an interplay

**9. L33: and commonly occur: something with grammer there**

*Changed to "frequent"*

"[..] and frequent in modern glacierised mountain area"

**10. L34-36: too many refs that refer to a very broad statement**

*We limited references to review studies and couple of regional examples:*

"frequent in modern glacierised mountain areas (Russell et al., 2007; Moore et al., 2009; Iribarren et al., 2015; Harrison et al., 2018; Nie et al., 2018; Carrivick and Tweed, 2019)."

**11. L38: of a moraine dam**

*Corrected*

**12. L44: in the case of Zack, I believe we will rather see the opposite with glacier thinning??**

*The lake, which is the source of GLOF, is located more than 3 km from the current ice margin, so we expect a similar or higher frequency as more water will be stored behind the glaciers. Thus, future monitoring will answer whether the GLOS will be observed more frequently but with lower discharge magnitude or less often but with higher discharge. We added the following explanation:*

"The first GLOF at Zackenberg was observed in 1996 and since then floods occurred every year or at the two-year interval (Kroon et al., 2017; Tomczyk and Ewertowski, 2020). The lake, which is the

source of GLOF, is located more than 3 km from the current ice margin, so we expect a similar or higher frequency as more water will be melting from glaciers and stored in the lake. Thus, future monitoring is needed to investigate whether the GLOFs will be observed more frequently but with lower discharge magnitude or less often but with higher discharge. *"*

**13. L48: related to a glacier lake…**

*corrected*

**14. L48: leaving behind serious… sounds jargon to me**

*Changed to "substantial":*

"On 6 August 2017, a flood event related to a glacier lake outburst affected the Zackenberg River (NE Greenland), leaving behind substantial geomorphological impacts on the riverbanks and channel morphology"

**15. L61-66: direct repetition of abstract.**

*Yes, as we wanted to emphasise potential applications of the presented dataset and stressed that only one of the proposed six applications was already shown in detail in our papers.*

**16. L71: suggest 'glacier-covered' instead of glaciated**

*Changed as suggested:*

"[…]its catchment covers 514 km$^2$, 20% of which is glacier-covered."

**17. L69: ZR could warrant an abbreviation**

*We would prefer to use the full name, as ZR could make text harder to read or be mistaken Zirconium*

**18. L74: check refs and GEM database. It is >200 to my knowledge.**

*We counted events with discharge > 100 m$^3$ s$^{-1}$ - There were 14 such events between 1996 and 2018, according to the GEM database. Please, also see the figure attached as a supplement.*

**19. L77: do Bendixen et al refer to ZR specifically? Check also Landegaard-Pedersen 2017 for the sediment part**

*Bendixen et al., 2017 investigated 121 deltas in south Greenland, but not Zackenberg. We added the reference to Ladegaard-Pedersen et al., 2017 and removed Bendixen et al., 2017, to keep only references related to Zackneberg valley:*

"Such events had an enormous impact on the riverscape geomorphology (Tomczyk and Ewertowski, 2020; Tomczyk et al., 2020), discharge and sediment transport (Hasholt et al., 2008; Søndergaard et al., 2015; Ladegaard-Pedersen et al., 2017), and delivery of nutrients and sediments into the fiord and delta development (Kroon et al., 2017)."

**20. Fig 1: Is it Young Sund or Young Sound in English? Unsure. What is the basis for the isolines? Acquisition date of ice margin extent? The bridge is referred to in the text but not in the map**

*We corrected the name to "Young Sound". The base for isolines is 0, and isolines labels were added to the map. The glacier cover is from 2001, and such information is now in the legend. We also provided a new panel (Fig. 1D) with a zoom of the study area – the location of the bridge and extent of the subsequent figures are now indicated in this new panel, as Fig. 1C was too small to fit all extent indicators.*

[Figure]

**Figure 1: Location of the study area (Reprinted from Journal of Hydrology, Vol 591, Tomczyk et al., Geomorphological impacts of a glacier lake outburst flood in the high arctic Zackenberg River, NE Greenland, 125300, Copyright (2020), with permission from Elsevier). Fig. 1D shows survey area with extent of Figures 2, 3, 6 and 7 indicated with boxes.**

**21. L91: what is medium-term in this context?**

*We meant several years (i.e. changes between 2014 and 2017) in contrast with short-term (i.e. changes over several days). Therefore, the text has been clarified as follow:*

"collect data that would enable quantifying medium-term (i.e., temporal scale of several years) changes in the river landscape."

**22. L91-95: repetition to the above?**

*We would prefer to keep this fragment as it explains the motivation of the surveys.*

**23. L97: delete: relatively long?**

*Deleted as suggested:*

covering a 2.1-km-long section of the river from the bridge to the delta.

**24. L98-99: delete. It comes later. No ref to TLS (wrong abbreviation used) necessary in my opinion if you don't use it. If at all in the discussion**

*The sentence was deleted as suggested.*

**25. L100-106: add brand of sensors**

*We added the brand of the sensor:*

The UAV was equipped with DJI 20MP, 1-inch size CMOS RGB sensor and a global shutter – camera model FC6310 (Table 1).

**26. Fig. 2: show in Fig 1 extent. Is it really 0.5 m you get from GoogleMaps? Doublecheck**

*We added the extent of the Figure 2 in Fig. 1. Yes, for this part of the Zackenberg Valley, high-resolution satellite imagery is available from the WorldView2 satellite (by Maxar, which is the current operator of the former DigitalGlobe constellation), which has 0.54 m GSD after pansharpening. Scene ID: 103001001ABA7E00, details of this scene can be found here: https://discover.digitalglobe.com/11608746-e421-11eb-ac2b-ee9402db8ad9*

**27. L123: what are good weather conditions? Specify or omit**

*Clarified as follow:*

"The weather condition for each day was good (i.e., no precipitation nor strong winds), and illumination conditions were sunny."

**28. L129: remove: 'so'**

*We hanged "so" to "therefore" to indicate a continuation of the previous sentence.*

"Therefore, it was a compromise between photogrammetric quality (i.e., the image network geometry), desired GSD, and rapidly changing flood conditions."

**29. L148: why? It is part of the station infrastructure. In 2017 the bridge was there, was it not possible to cross? I am surprised and a bit doubtful by the lack of GCPs**

*We did not have access to cm-accuracy survey equipment; only dm-scale GPS available was available. During the flood, the bridge was unpassable as the water level was too high to reach it. We added the clarification*

(we did not have access to cm-accuracy survey equipment, and it was not possible to cross the river during the flood, because of the high water level),

**30. L150: This is all permafrost, even large boulders may move due to creep, definitely between years. I am doubtful that the accuracy can be achieved down to say 4 mm (Tab1) if no GCPs are taken. Even if we refer to the 0.12 m to 0.15 m this may be within the moving conditions of the terrain over 4 summer seasons if you use the COWI DEM. This warrants at least discussion.**

*Each of the individual camera positions (i.e., every image captured) have 3D position coordinates originated from the onboard UAV GNSS system. These coordinates gave absolute (i.e., external) orientation, further constrained by control points (CPs) obtained from high-resolution 2014 data. We selected CPs located in flat areas which are less likely to move, including stable features like buildings of Zackenberg Research Station. The CPs were used to ensure that absolute accuracy (i.e., the geometry of the reconstructed scene in relation to the outside world) is correct (RMSE on checkpoints was between 0.12 m and 0.15 m). As stated in Table 1 - The 4 mm mentioned by the Reviewer is the mean point coordinate precision of surveys – as indicated in Table 1 and the text. It shows the stability of the reconstructed point cloud and can be used to investigate the spatial uncertainty of each of the point clouds. Therefore, datasets processed by us are characterised by two types of accuracy (as indicated in Table 1):*

1) *The absolute accuracy (i.e., the position of all surveys in relation to the outside world) is on a dm-level*
2) *The relative accuracy (i.e., stability of each individual point cloud) is on cm-level*

*Such a level of accuracy is sufficient for most applications, including modelling, because the relative (i.e. internal) accuracy is much higher than absolute accuracy. It is also sufficient for change detection because observed changes were up to 10 m of lateral erosion, so 100 times larger than RMSE on external checkpoints. If better relative accuracy is necessary in the future for some additional applications, co-alignment of UAV time-series can provide better relative (i.e., internal survey-to-survey) accuracy than the classic approach of individual SfM processing of each survey using GCP - as demonstrated in several studies (e.g., Cook and Dietze, 2019; de Haas et al., 2021), Therefore, we provided also unprocessed images so the potential user can perform their own SfM processing. We added the following fragment to Section 3.1 "Structure-from-Motion processing", and to Section 4 "Quality assessment and known limitations"):*

Section 3.1

The external orientation of the reconstructed scene was established using coordinates of each camera position obtained from the onboard GNSS system. To further constrain the geometry of the scene, additional control points were used (CPs). […] CPs were then generated post-survey using previous UAV dataset from 2014 (COWI, 2015). In total, 100 points were selected, located mostly on stable, flat boulders, which were easy to identify in the images. CPs were distributed on level terrain to minimize the impact of potential permafrost creep.

Section 4

Although such values are higher than the GSD of all datasets (between 0.018 and 0.028 m), such magnitude of errors was considered acceptable for the quantification and mapping of landscape changes, especially as between 5 August and 8 August the resultant lateral erosion of riverbanks from the flood reached almost 10 m in some sections, (see Tomczyk et al., 2020 for details), therefore the observed changes were up 100 times larger than RMSE. If necessary, lower values of absolute accuracy can be achieved in the future if additional ground control points are surveyed using cm-accuracy survey equipment. Moreover, if better relative accuracy (i.e. survey-to-survey accuracy) is necessary in the future monitoring applications, co-alignment of UAV time-series can provide better relative accuracy than the classic approach of individual SfM processing of each survey using GCP - as demonstrated in several studies (e.g., Feurer and Vinatier, 2018; Cook and Dietze, 2019; de Haas et al., 2021). Therefore, we provided also unprocessed images so the potential user can perform their own SfM processing.

**31. L158: which units?**

*The units were pixels. Information added as follow:*

"[…]with low tie point reprojection errors from 0.28 to 0.44 pixels,"

**32. L173: ground-truthing**

*Corrected as suggested*

**33. L173: vector were vectorized… reword**

*Modified to shapefiles as follow:*

Final shapefile datasets were vectorised on-screen in ArcMap 10.6 software.

**34. Fig 3: the color scale of a and e is not useful: use contours and descrete colours as well as contour labels. Also very strange: after the flood the elevation of the orographically left river bank is in the order of 13 m higher than the right river bank?? I have some doubts that this is realistic undercut, also when I look at 3g. if that was the cassse, the yellow colours would likely disappear in 3e and just the plateau remains. Mark extent in fig 1. Scarp and drainage is strange to me here. What is the water body in the upper left corner of the fig? Does a map with different colours related to the slope qualify as 'geomorphological mapping'?**

*We changed the scale to discrete colours and added contours with labels. Fig. 3 shows a steep riverbank that is indeed 13 m high in this place and was severely undercut during the flood and mid-channel bar, which was covered by water during the flood but re-emerged after the water level dropped. We hope that indication of the location of this river section in Fig.2. will be helpful to orientate the reader with the overall geomorphological situation. We also added Fig. 7 – it shows photographs of this section of the river and other undercut sections. Yes, this is a geomorphological map as it contains several different geomorphological features – floodplain, scrap, slope (divided into different categories).*

[Figure]

**Figure 3: Examples of delivered dataset illustrating before and after the flood situation: (a, e) digital elevation model; (b, f) hillshade model; (c, g) orthomosaics; (d, h) results of geomorphological mapping**

[Figure]

**Figure 7: Examples of steep and undercut riverbanks.**

**35. L189: how do you relate this accuracy to potential permafrost creep?**

*As mentioned before, external orientation was based on onboard GNSS coordinates constrained by CPs. CPs were located on stable boulders in level ground, so the impact of the permafrost creep was minimized.*

**36. L194: by then they will have moved again to some extent I fear. It is necessary in such an environment to have DGPS GCPs for each survey if such high accuracy should be obtained.**

*As mentioned in response to general comments, our data cover stable grounds (e.g., buildings of the research station and fragments of marine terraces), so there will be enough stable points to co-register future surveys. Moreover, de Haas et al. (2021) demonstrated that even in such an unstable environment as debris flow torrent, the co-alignment approach gave relative accuracy of change detection better by a factor 3 than the classical approach with individually processed surveys with GCPs. Therefore, we provided also unprocessed images so the potential user can perform their own SfM processing.*

**37. L195-205: please make it very clear what is precision within the 3 DEMs and what is the absolute precision.**

*We added the following clarification:*

The quality of the presented datasets was assessed in relation to the outside world (i.e., external or absolute accuracy) and in relation to each survey (internal precision). […] The external accuracy was estimated based on root-mean-square errors (RMSE) and standard deviations (SD) of errors on checkpoints, which were between 0.12 and 0.15 m (Table 1). The maximum external error for two checkpoints was -0.4 m and 0.4 m (Fig. 5).

*and*

The internal quality of the reconstructed scenes was based on tie point precision. […] The internal accuracy of each survey was assessed based on the mean point precision estimates, which varied from 4 to 6 mm for the horizontal component and from 11 to 15 mm for the vertical one (Table 1)

**38. Fig 4: the colour scale is unacceptable. Have contours or discrete ones. I cant see anything here. Also indicate where you are in an overview fig**

*The Colour scale was changed to discrete classes. The location of the figure is now indicated in Fig 1, as a Fig 1D.*

[Figure]

**Figure 4: Precision estimates for X, Y, Z coordinates of tie points. Location of the studied river section is presented in Fig. 1D.**

**39. Fig 5: same as above. Unclear what is shown. which tie point centroid? So here we have errors of up to 0.4 m?**

*It shows the distribution of errors on checkpoints and control points. Such information is used to verify if there are systematic errors with reconstructed geometry, such as doming or dishing of the model (please, see James et al. (2020) for details. Yes, individual external errors measured versus checkpoints were up to -0.4 m and 0.4 m for two of the checkpoints for the 8 August survey. We added the following clarification:*

The external accuracy was estimated based on root-mean-square errors (RMSE) and standard deviations (SD) of errors on checkpoints, which were between 0.12 and 0.15 m (Table 1). The maximum external error for two of the checkpoints was -0.4 m and 0.4 m.

*And*

Figure 6: Spatial distribution of errors on control and checkpoints: (a) Z-error against radial distance from the tie point cloud centroid (i.e. from the centre of the reconstructed scene). The distribution of errors along a straight line (indicated here also as "modelled constant") suggests that no systematic errors such as doming or dishing were observed in the reconstructed scenes (see James et al., 2020 for details); (b) Z-error by colour in plan view (X, Y are distanced from tie points centroid). Note: Each row shows an individual survey.

**40. L216-220: it is really unclear why this is relevant there. Stick to topics that occur**

*Shortened as suggested:*

In general, the interpretation of riverbank conditions can be tampered by vegetation cover and/or bank undercutting (Niedzielski et al., 2016; Hemmelder et al., 2018). While vegetation cover is usually not a problem in the case of Arctic rivers, other obstacles (e.g., shadows, infrastructure) might prevent the direct measurements of the bank's heights. In the case of the presented dataset, some sections of riverbanks were steep, near-vertical, before the flood.

**41. L224: do you have pictures of this undercut? It is a bit hard to believe to me**

*We provided additional figure 7 with ground-based photos of examples of the undercuts produced by the 2017 flood.*

[Figure]

**Figure 7: Examples of steep and undercut riverbanks.**

**42. L235: yellowish is jargon I believe**

*Changed to "yellow"*

"indicated by the lack of transparency and the yellow or brown colours of water in the orthomosaics."

**43. L240: how can you assess the character of water flow from orthophotos??**

*It can be assessed by the character of the water surface, e.g., gentle and level surface in sections where the water flow was relatively slow vs surfaces covered with small waves, which indicated rapid water flow. We added Fig. 8 demonstrating different types of water surfaces attached in the supplement.*

[Figure]

**Figure 8: Different character of water surfaces: (a) stagnant and slow flowing water; (b) moderate flow rate; (c) rapid, turbulent water flow.**

**44. L287: precision? Typo**

*corrected*

**45. Tab 2: why not showing the vectors in all maps (bridge, track). Is it really a 4x4 track??**

*We added the bridge and buildings into the Figure 1D (overview map). It is a track used by Argo all-terrain-vehicle. Other figures present detailed examples of small areas and these features are not in their extents.*

**46. L295-317: largely another repetition**

*This section contains a more detailed description of applications that we believe are the most interesting, together with references to some studies on modelling, which can guide future approaches. This information was not presented earlier, merely mentioned at the end of the introduction and in the abstract (which is often read separately from the whole paper). However, we modified this section as follow (changes also include suggestions from Referees #1 and #2)*

1) Quantification, monitoring and modelling of geomorphological impacts of glacier lake outburst flood – the presented dataset was meant to quantify changes related to the 2017 GLOF (see Tomczyk and Ewertowski, 2020; Tomczyk et al., 2020); however, these studies only described the immediate impacts of a single flood event. An example of geomorphic change detection is presented in Fig. 9, demonstrating the acceleration of debris flows resulting from sediment entrainment at the base of the river bans by floodwater. Overall, the observed changes were spatially variable – erosion dominated along steep banks as expected; however, understanding of differences in erosion rates between sites requires further studies, which will consider differences in lithology as well as modelling of water flow to investigate potential erosion forces in relation to channel characteristics. The first GLOF at Zackenberg was observed in 1996 and since then floods occurred every year or at the two-year interval (Kroon et al., 2017; Tomczyk and Ewertowski, 2020). The lake, which is the source of GLOF, is located more than 3 km from the current ice margin, so we expect a similar or higher frequency as more water will be melting from glaciers and stored in the lake. Thus, future monitoring is needed to investigate whether the GLOFs will be observed more frequently but with lower discharge magnitude or less often but with higher discharge.

2) As the high-magnitude low-frequency events are typically rare and difficult to predict, our understanding of the quantitative aspect of geomorphological changes related to them remains limited compared to the "normal" processes (Tamminga et al., 2015b). These arise particularly from difficulties in collecting high-resolution data before and after these innately unpredictable and rare flood events. However, investigation into the geomorphological response of river morphology to "extreme" events is key to understanding the evolution of river morphology and being crucial from the standpoint of river modelling and monitoring (Tamminga et al., 2015a; Tamminga et al., 2015b). Moreover, the relationship between the magnitude of the flood and geomorphological effects is not fully understood. For example, in the case of Zackenberg River, immediate (2-days) lateral erosion compared to three-year erosion was spatially very diversified. In some sections, immediate lateral erosion after the 2017 flood reached up to 10 m, whereas the same section was stable between 2014 and 2017, even though higher peak discharges characterised 2015 and 2016 GLOFs than 2017 GLOF (Tomczyk et al., 2020). Further process-based studies are necessary to observe and model links between the magnitude of a flood and the severity of erosion. It is especially important in periglacial landscapes where lateral bank erosion can be responsible for delivering a large quantity of organic matter and widespread changes in ecosystems. especially combined with other weather extreme events (see Christensen et al., 2021). Using the provided dataset as a baseline for the monitoring of future changes, it should be possible to quantify the difference between geomorphological effects of "normal" (i.e., high-frequency, low-magnitude) processes on the one hand, and extreme (i.e., low-frequency, high-magnitude) events on the other. Also, by linking the intensity of a geomorphological response to hydrological data about flood characteristics, it should be

possible to improve modelling routines (cf. Carrivick, 2007a, b; Carrivick et al., 2011; Guan et al., 2015; Staines and Carrivick, 2015).

**References:**

Carrivick, J. L.: Modelling coupled hydraulics and sediment transport of a high-magnitude flood and associated landscape change, Ann Glaciol, 45, 143-154, doi:10.3189/172756407782282480, 2007a.

Carrivick, J. L.: Hydrodynamics and geomorphic work of jökulhlaups (glacial outburst floods) from Kverkfjöll volcano, Iceland, Hydrol Process, 21, 725-740, doi:10.1002/hyp.6248, 2007b.

Carrivick, J. L., Jones, R., and Keevil, G.: Experimental insights on geomorphological processes within dam break outburst floods, J Hydrol, 408, 153-163, doi:https://doi.org/10.1016/j.jhydrol.2011.07.037, 2011.

Carrivick, J. L., Turner, A. G. D., Russell, A. J., Ingeman-Nielsen, T., and Yde, J. C.: Outburst flood evolution at Russell Glacier, western Greenland: effects of a bedrock channel cascade with intermediary lakes, Quaternary Sci Rev, 67, 39-58, doi:https://doi.org/10.1016/j.quascirev.2013.01.023, 2013.

Carrivick, J. L., Tweed, F. S., Ng, F., Quincey, D. J., Mallalieu, J., Ingeman-Nielsen, T., Mikkelsen, A. B., Palmer, S. J., Yde, J. C., Homer, R., Russell, A. J., and Hubbard, A.: Ice-Dammed Lake Drainage Evolution at Russell Glacier, West Greenland, Frontiers in Earth Science, 5, doi:10.3389/feart.2017.00100, 2017.

Carrivick, J. L., and Tweed, F. S.: A review of glacier outburst floods in Iceland and Greenland with a megafloods perspective, Earth-Sci Rev, 196, 102876, doi:https://doi.org/10.1016/j.earscirev.2019.102876, 2019.

Chandler, B. M. P., Lovell, H., Boston, C. M., Lukas, S., Barr, I. D., Benediktsson, Í. Ö., Benn, D. I., Clark, C. D., Darvill, C. M., Evans, D. J. A., Ewertowski, M. W., Loibl, D., Margold, M., Otto, J.-C., Roberts, D. H., Stokes, C. R., Storrar, R. D., and Stroeven, A. P.: Glacial geomorphological mapping: A review of approaches and frameworks for best practice, Earth-Sci Rev, 185, 806-846, doi:10.1016/j.earscirev.2018.07.015, 2018.

Christensen, T. R., Lund, M., Skov, K., Abermann, J., López-Blanco, E., Scheller, J., Scheel, M., Jackowicz-Korczynski, M., Langley, K., Murphy, M. J., and Mastepanov, M.: Multiple Ecosystem

Effects of Extreme Weather Events in the Arctic, Ecosystems, 24, 122-136, doi:10.1007/s10021-020-00507-6, 2021.

Cook, K. L., and Dietze, M.: Short Communication: A simple workflow for robust low-cost UAV-derived change detection without ground control points, Earth Surface Dynamics, 7, 1009-1017, doi:10.5194/esurf-7-1009-2019, 2019.

Mapping Greenland's Zackenberg Research Station: https://www.sensefly.com/app/uploads/2017/11/eBee_saves_day_mapping_greenlands_zackenberg_research_station.pdf, access: 20.11, 2015.

Dawson, A. G.: Glacier-dammed lake investigations in the Hullet Lake area, South Greenland, Museum Tusculanum Press, 1983.

de Haas, T., Nijland, W., McArdell, B. W., and Kalthof, M. W. M. L.: Case Report: Optimization of Topographic Change Detection With UAV Structure-From-Motion Photogrammetry Through Survey Co-Alignment, Frontiers in Remote Sensing, 2, doi:10.3389/frsen.2021.626810, 2021.

Feurer, D., and Vinatier, F.: Joining multi-epoch archival aerial images in a single SfM block allows 3-D change detection with almost exclusively image information, ISPRS Journal of Photogrammetry and Remote Sensing, 146, 495-506, doi:https://doi.org/10.1016/j.isprsjprs.2018.10.016, 2018.

Furuya, M., and Wahr, J. M.: Water level changes at an ice-dammed lake in west Greenland inferred from InSAR data, Geophysical Research Letters, 32, doi:https://doi.org/10.1029/2005GL023458, 2005.

Grinsted, A., Hvidberg, C. S., Campos, N., and Dahl-Jensen, D.: Periodic outburst floods from an ice-dammed lake in East Greenland, Scientific reports, 7, 9966, doi:10.1038/s41598-017-07960-9, 2017.

Guan, M., Wright, N. G., Sleigh, P. A., and Carrivick, J. L.: Assessment of hydro-morphodynamic modelling and geomorphological impacts of a sediment-charged jökulhlaup, at Sólheimajökull, Iceland, J Hydrol, 530, 336-349, doi:https://doi.org/10.1016/j.jhydrol.2015.09.062, 2015.

Harrison, S., Kargel, J. S., Huggel, C., Reynolds, J., Shugar, D. H., Betts, R. A., Emmer, A., Glasser, N., Haritashya, U. K., Klimeš, J., Reinhardt, L., Schaub, Y., Wiltshire, A., Regmi, D., and Vilímek, V.: Climate change and the global pattern of moraine-dammed glacial lake outburst floods, The Cryosphere, 12, 1195-1209, doi:10.5194/tc-12-1195-2018, 2018.

Hasholt, B., Mernild, S. H., Sigsgaard, C., Elberling, B., Petersen, D., Jakobsen, B. H., Hansen, B. U., Hinkler, J., and Søgaard, H.: Hydrology and Transport of Sediment and Solutes at Zackenberg, in: Advances in Ecological Research, Academic Press, 197-221, 2008.

Hasholt, B., van As, D., Mikkelsen, A. B., Mernild, S. H., and Yde, J. C.: Observed sediment and solute transport from the Kangerlussuaq sector of the Greenland Ice Sheet (2006–2016), Arctic, Antarctic, and Alpine Research, 50, S100009, doi:10.1080/15230430.2018.1433789, 2018.

Hemmelder, S., Marra, W., Markies, H., and De Jong, S. M.: Monitoring river morphology & bank erosion using UAV imagery – A case study of the river Buëch, Hautes-Alpes, France, International Journal of Applied Earth Observation and Geoinformation, 73, 428-437, doi:https://doi.org/10.1016/j.jag.2018.07.016, 2018.

Iribarren, P., Mackintosh, A., and Norton, K. P.: Hazardous processes and events from glacier and permafrost areas: lessons from the Chilean and Argentinean Andes, Earth Surf Proc Land, 40, 2-21, doi:10.1002/esp.3524, 2015.

James, M. R., Chandler, J. H., Eltner, A., Fraser, C., Miller, P. E., Mills, J. P., Noble, T., Robson, S., and Lane, S. N.: Guidelines on the use of structure-from-motion photogrammetry in geomorphic research, Earth Surf Proc Land, 44, 2081-2084, doi:10.1002/esp.4637, 2019.

James, M. R., Antoniazza, G., Robson, S., and Lane, S. N.: Mitigating systematic error in topographic models for geomorphic change detection: accuracy, precision and considerations beyond off-nadir imagery, Earth Surf Proc Land, 45, 2251-2271, doi:https://doi.org/10.1002/esp.4878, 2020.

Kroon, A., Abermann, J., Bendixen, M., Lund, M., Sigsgaard, C., Skov, K., and Hansen, B. U.: Deltas, freshwater discharge, and waves along the Young Sound, NE Greenland, Ambio, 46, 132-145, doi:10.1007/s13280-016-0869-3, 2017.

Ladegaard-Pedersen, P., Sigsgaard, C., Kroon, A., Abermann, J., Skov, K., and Elberling, B.: Suspended sediment in a high-Arctic river: An appraisal of flux estimation methods, Sci Total Environ, 580, 582-592, doi:https://doi.org/10.1016/j.scitotenv.2016.12.006, 2017.

Mayer, C., and Schuler, T. V.: Breaching of an ice dam at Qorlortossup tasia, south Greenland, Ann Glaciol, 42, 297-302, doi:10.3189/172756405781812989, 2005.

Moore, R. D., Fleming, S. W., Menounos, B., Wheate, R., Fountain, A., Stahl, K., Holm, K., and Jakob, M.: Glacier change in western North America: influences on hydrology, geomorphic hazards and water quality, Hydrol Process, 23, 42-61, doi:Doi 10.1002/Hyp.7162, 2009.

Nie, Y., Liu, Q., Wang, J., Zhang, Y., Sheng, Y., and Liu, S.: An inventory of historical glacial lake outburst floods in the Himalayas based on remote sensing observations and geomorphological analysis, Geomorphology, 308, 91-106, 2018.

Niedzielski, T., Witek, M., and Spallek, W.: Observing river stages using unmanned aerial vehicles, Hydrol. Earth Syst. Sci., 20, 3193-3205, doi:10.5194/hess-20-3193-2016, 2016.

Russell, A. J., Gregory, A. R., Large, A. R. G., Fleisher, P. J., and Harris, T. D.: Tunnel channel formation during the November 1996 jokulhlaup, Skeioararjokull, Iceland, Ann Glaciol, 45, 95-103, 2007.

Russell, A. J.: Jökulhlaup (ice-dammed lake outburst flood) impact within a valley-confined sandur subject to backwater conditions, Kangerlussuaq, West Greenland, Sediment Geol, 215, 33-49, doi:https://doi.org/10.1016/j.sedgeo.2008.06.011, 2009.

Russell, A. J., Carrivick, J. L., Ingeman-Nielsen, T., Yde, J. C., and Williams, M.: A new cycle of jökulhlaups at Russell Glacier, Kangerlussuaq, West Greenland, J Glaciol, 57, 238-246, doi:10.3189/002214311796405997, 2011.

Søndergaard, J., Tamstorf, M., Elberling, B., Larsen, M. M., Mylius, M. R., Lund, M., Abermann, J., and Rigét, F.: Mercury exports from a High-Arctic river basin in Northeast Greenland (74°N) largely controlled by glacial lake outburst floods, Sci Total Environ, 514, 83-91, doi:https://doi.org/10.1016/j.scitotenv.2015.01.097, 2015.

Staines, K. E. H., and Carrivick, J. L.: Geomorphological impact and morphodynamic effects on flow conveyance of the 1999 jökulhlaup at sólheimajökull, Iceland, Earth Surf Proc Land, 40, 1401-1416, doi:10.1002/esp.3750, 2015.

Tamminga, A., Hugenholtz, C., Eaton, B., and Lapointe, M.: Hyperspatial remote sensing of channel reach morphology and hydraulic fish habitat using an unmanned aerial vehicle (UAV): A first assessment in the context of river research and management, River Res Appl, 31, 379-391, 2015a.

Tamminga, A. D., Eaton, B. C., and Hugenholtz, C. H.: UAS-based remote sensing of fluvial change following an extreme flood event, Earth Surf Proc Land, 40, 1464-1476, 2015b.

Tomczyk, A. M., and Ewertowski, M. W.: UAV-based remote sensing of immediate changes in geomorphology following a glacial lake outburst flood at the Zackenberg river, northeast Greenland, J Maps, 16, 86-100, doi:10.1080/17445647.2020.1749146, 2020.

Tomczyk, A. M., Ewertowski, M. W., and Carrivick, J. L.: Geomorphological impacts of a glacier lake outburst flood in the high arctic Zackenberg River, NE Greenland, J Hydrol, 591, 125300, doi:10.1016/j.jhydrol.2020.125300, 2020.

Weidick, A., and Citterio, M.: The ice-dammed lake Isvand, West Greenland, has lost its water, J Glaciol, 57, 186-188, doi:10.3189/002214311795306600, 2011.

Yde, J. C., Žárský, J. D., Kohler, T. J., Knudsen, N. T., Gillespie, M. K., and Stibal, M.: Kuannersuit Glacier revisited: Constraining ice dynamics, landform formations and glaciomorphological changes in the early quiescent phase following the 1995–98 surge event, Geomorphology, 330, 89-99, doi:https://doi.org/10.1016/j.geomorph.2019.01.012, 2019.